# Effect of the Addition of Chia Seed Gel as Egg Replacer and Storage Time on the Quality of Pork Patties

**DOI:** 10.3390/foods10081744

**Published:** 2021-07-29

**Authors:** Mirosława Karpińska-Tymoszczyk, Marzena Danowska-Oziewicz, Anna Draszanowska

**Affiliations:** Department of Human Nutrition, Faculty of Food Science, University of Warmia and Mazury in Olsztyn, 10-718 Olsztyn, Poland; marzena.danowska@uwm.edu.pl (M.D.-O.); anna.draszanowska@uwm.edu.pl (A.D.)

**Keywords:** pork patties, chia seeds, storage, physiochemical properties, sensory quality

## Abstract

Two types of patties were prepared: control and with chia seeds gel instead of beaten egg. The patties were cooked in the steam-convection oven, vacuum packed and stored at 4 °C. The pork patties with chia addition were characterized by similar water activity and pH values to the control samples. They showed lower values of the b* colour parameter as well as colour saturation (C*) and hue angle values (h°) on the cross-section and lower values of colour parameters L*, a* and b* and C* on the surface than the controls. The addition of chia seeds improved the texture parameters of the tested products. Pork patties with chia seeds were softer and showed better chewiness than the control samples. Chia slowed down oxidative changes in pork patties during storage. The use of 8.0% addition of chia seeds was only slightly noticeable in taste of the pork patties and these samples received similar overall quality scores as control samples.

## 1. Introduction

Chia (*Salvia hispanica* L.) is an annual herbaceous plant that belongs to the *Lamiaceae* family. Chia seeds are gaining interest in food science and industry as it was reported they contain a considerable amount of essential fatty acids in the oil, dominated by α-linolenic acid, as well as low share of saturated fatty acids [1,2]. Another interesting feature of chia seeds is a presence of biologically active compounds, particularly polyphenols such as gallic, caffeic, chlorogenic, cinnamic and ferulic acids, quercetin, kaempferol, epicatechin, rutin, apigenin and p-coumaric acid [3,4] as well as other antioxidants and dietary fiber [4]. Because of pro-healthy importance and functional properties, the seeds could be utilized in a production of functional foods [5]. Chia seeds are characterized by an advantageous content of phenolic compounds which show antioxidant properties, and which are believed to have cardiac and hepatic protective effect as well as anti-ageing and anti-carcinogenic characteristics [6]. Several studies showed that the chia seeds, and consequently their bioactive compounds, presence in the diet was related to the lower occurrence of cardiovascular disease and hepatoprotective effect [7], and also positive action against plasma oxidative stress and obesity related diseases [8].

Addition of chia seeds increases nutritional and pro-healthy value of food products as well as can also improve technological features like water-holding capacity, water absorption capacity, emulsifying activity or gelling capacity [9]. It was reported that the amount and method of ingredients incorporation to meat systems, especially those of gel/emulsion nature, influence composition and organoleptic characteristics of final products, and also determine their application in production of novel or reformulated meat products [10].

The objective of this study was to evaluate how eggs replacing in pork patties with chia seeds gel would affect the chemical composition, pH value, colour and texture parameters, sensory quality and lipid oxidation compared to full egg recipe (control) during chilled storage.

## 2. Materials and Methods

### 2.1. Materials

The raw material was pork shoulder purchased from a local meat supplier in the Region of Warmia and Mazury (Poland). Meat originated from the carcasses of animals slaughtered at the age of 5.5 months, with body weight 115–120 kg, 24 h after slaughter. Chia seeds were of Peru/Bolivia origin and were purchased at a discount store in Olsztyn (Poland). The other components used in the manufacturing of pork patties were eggs, wheat roll, pure natural salt, black pepper and onion.

### 2.2. Samples Preparation

The composition of pork patties was established on the basis of the preliminary study that included preparation of patties using different proportions of components and sensory evaluation of products by a trained panel. Pork patties consisted of pork shoulder (70%), wheat roll soaked in water (14.0%), fried onion (8.0%) and beaten egg (8.0%) in the control samples or chia seed gel (8.0%) in the samples with chia. The other ingredients: salt (0.5%) and pepper (0.03%) were added in relation to total mass weight. Chia seeds were mixed with cold water in proportion 1:15 and mixture was stored for 30 min. The onion was diced and fried in small amount of rapeseed oil. The meat was minced twice in the Mesko-AGD KU2-4E grinder (Skarżysko-Kamienna, Poland) with a 5 mm grinding plate and mixed in the Mankiewicz SP-100A-B mixer (Radzionków, Poland) for 10 min. Then bread roll soaked in water and fried onion were added and all components were mixed for the next 10 min to obtain a uniform mass. The whole mass was divided into two parts and to one part beaten egg, salt and pepper while to the second part chia seed gel, salt and pepper were added, followed by mixing both samples for 20 min in the meat mixer. Three separate batches were prepared as three technological replicates of the experiment.

The 80 g portions were weighed from the prepared mass and formed into patties (7 cm diameter, 1 cm thickness). Ninety patties were prepared from each treatment (30 × 3 replicates).

### 2.3. Cooking

The patties were placed on the oven tray (fifteen patties at a time) and cooked in the Rational SCCWE-101 steam-convection oven (Landsberg am Lech, Germany), using hot air of 140 °C and relative humidity 50%. The heat treatment was performed to the internal end-point temperature of 72 °C. The temperature was measured with the oven temperature probe.

### 2.4. Product Storage

The cooked patties were allowed to cool down to room temperature and vacuum packed—three patties in one bag—with the use of the Edesa VAC-20 DT chamber vacuum sealer (Barcelona, Spain). The bags were made from multilayer PA/PE barrier plastic foil with a thickness of 52 µm (Hendi, Lamprechtshausen, Austria). The samples were stored in a laboratory refrigerator (MediLine, LKexv 3600, Liebherr, Austria) at a constant temperature of 4 °C and analyzed after 24 h (day 0) and after 6 and 12 days of refrigerated storage.

### 2.5. Proximate Composition

The moisture content of patties was determined by drying to constant weight [11]. Protein content was determined by the Kjeldahl method [12], and fat content—by the Soxhlet method [13]. All analyses were performed in three replications.

### 2.6. Water Activity

An comminuted sample was placed into a measuring container and three measurements were performed at 20 °C for each treatment at days 0, 6 and 12 of the experiment. The water activity analyser was AWC 203-C (Novasina, Pfäffikon, Switzerland). Prior to evaluation, the samples were taken out from the refrigerator to reach an ambient temperature.

### 2.7. pH Measurement

Ten grams of each treatment was homogenized with 50 mL of distilled water using the HO 4A homogenizer (Edmund Bühler GmbH, Hechingen, Germany) for 2 min at 6000 rpm. The pH value of the samples was measured with the Hanna Instruments 210 pH meter (Romania) calibrated with pH 7 and pH 4 buffers before measurements. For each formulation three measurements were performed.

### 2.8. Lipid Oxidation

The TBARS value of experimental patties was determined by the method described in Salih et al. [14]. Ground samples of 10 g each were homogenized with 34.25 mL of cold (4 °C) 4% perchloric acid and 0.75 mL of an alcohol solution of butylhydroxytoluene (0.01%) using the HO 4A homogenizer (Edmund Bühler GmbH, Hechingen, Germany) at 4000 rpm for 2 min. The homogenate was passed through a Whatman 1 filter paper directly into a 50 mL measuring flask. The filtrate was made up to a volume of 50 mL by rinsing the sediment on filter paper with cold (4 °C) 4% perchloric acid. An aliquot of 5 mL sample was transferred to a test tube and 5 mL of 0.02 M aqueous solution of 2-thiobarbituric acid (TBA reagent) was added. Test tubes were capped with stoppers and heated in a boiling water bath for 60 min. After heating, test tubes were cooled for 10 min under cold running water. Absorbance was measured at a wavelength of 532 nm with the Optizen POP UV/VIS spectrophotometer (Metasys Co. Ltd., Daejeon, South Korea) and read against the blank sample containing 5 mL of 4% perchloric acid and 5 mL of the TBA reagent. For each formulation three measurements were performed. The TBARS value was calculated according to the equation presented below (1) and expressed in mg of malondialdehyde (MDA) per 1 kg of the product [15]:TBARS [mg MDA/kg] = A * K,(1)
where: A—absorbance of the analyzed sample, K—conversion factor of 5.5

### 2.9. Texture Profile Analysis (TPA)

The texture attributes of pork patties were analyzed by compressing 1 × 1 × 1 cm meat samples in two-cycle test to 50% of initial height [16]. TPA was performed in using a texture analyser TA.XT plus (Stable Micro Systems Ltd., Godalming, UK) with a 50 kg load cell. The analyser was equipped with the P/100 compression platen and the HDP/90 heavy duty platform. The crosshead speed was set at 5 mm/s. The determinations were performed at room temperature (20 ± 1 °C) and carried out in 20 replications. The following parameters were measured: hardness, chewiness, cohesiveness, springiness.

### 2.10. Colour

Colour parameters of patties were measured in CIE Lab (L*, a*, b*) colour space system with the use of CR-400 Chroma Meter (Konica Minolta, Osaka, Japan) equipped with a measurement port of 8 mm diameter. Before measurements the analyser was calibrated with the use of white tile supplied by the manufacturer. The colour parameters (L*-lightness, a*-redness, b*-yellowness) were determined using D65 illuminant and standard observer 2°. The colour saturation—chroma (C*) and hue angle (h°) were calculated according to the formulas:(2)$$C^{*}=\sqrt{a^{*2}+b^{*2}}$$
(3)$$h^{{}^{\circ}\;}=\mathrm{arctg}\;\left(\frac{b^{*}}{a^{*}}\right)$$
where: C*—colour saturation; h°—hue angle; a*—redness; b*—yellowness

Measurements were made 30 min after the packages had been opened. Colour was measured at three randomly selected locations on the cross-section and on the surface of three randomly selected patties of each formulation.

### 2.11. Sensory Quality

Patties immediately after taking out of the package were warmed up to 37 °C for 10 min in the steam-convection oven before the sensory evaluation. Six patties of each formulation were cut in half. Samples were coded with three digits numbers and served in random order to avoid carry-over effects. Encoded samples were presented to the panellists on identical white china plates in two separate sets during sensory session. Water and bread were available to the panellists during evaluation to clean a palate between samples. The evaluation was performed in a sensory analysis laboratory at ambient temperature. The team of twelve panellists (6 females and 6 males) was composed of students and employees of the Department of Human Nutrition with confirmed sensory sensitivity according to the ISO standard [17] and considerable experience in the sensory testing of meat products. Before evaluation the panellists were trained in two separated sessions using pork patties as reference material. The products were scored using a 10-point numerical-interval scale [18]. The sensory evaluation consisted of the assessment of appearance (1—undesirable; 10—very desirable), intensity of chia aroma (1—not detectable; 10—very strong), aroma acceptability (1—undesirable; 10—very desirable), juiciness (1—dry; 10—very juicy), tenderness (1—tough; 10—very tender), intensity of chia flavour (1—not detectable; 10—very strong), flavour acceptability (1—undesirable; 10—very desirable) and overall quality (1—very low; very high).

### 2.12. Statistical Analysis

The experiment was carried out in a complete randomized design with three replicates. The data was tested by F-test (ANOVA). Significant differences between samples and storage time were determined by the Tukey’s test, at *p* < 0.05 using STATISTICA software (TIBCO Software Inc. Tulsa, OK, USA). Results of analyses were combined and presented as mean values with standard deviation (SD).

## 3. Results and Discussion

### 3.1. Proximate Composition, Water Activity, pH

The chemical composition of the analyzed patties is shown in Table 1. The moisture content was significantly higher in the patties with the addition of chia seeds (70.63%) than in the control sample (69.15%), while they contained a lower amount of protein (18.10%) and fat (8.47%) than the control samples (20.27% and 9.08%, resp.).

Water activity in foodstuffs informs about the degree of link between water molecules and food ingredients and its value reflects the availability of water for chemical and biochemical reactions and microorganism growth, which determines the storage stability of food [19]. For most microorganisms, the optimal level of water activity for their development is 0.990–0.995. The minimum value of water activity necessary for the growth of individual groups of microorganism is 0.9 for most bacteria, 0.8 for most yeast and 0.7 for most mold [20]. Water activity in the control samples was similar to that in the samples with addition of chia seeds and no effect of storage time (*p* > 0.05) was shown on the value of this parameter (Table 1).

The pH value and microorganisms activities are closely related [21]. The study shows that pH values of the pork patties with chia seeds were lower than pH of the control samples, and although the differences were statistically significant they have no real practical importance (Table 1). Despite the pH changes observed in the pork patties as the effect of storage were small (lower than 0.2 pH units), they were significant. Pintado et al. [22] also showed small changes of pH values in frankfurters made with addition of olive oil-in-water emulsion gel prepared with chia flour during refrigerated storage. According to the research of Fernández-López et al. [23], pH and water activity of frankfurters were not altered by the incorporation of chia seeds and flour. Similar observations were made by Paula et al. [24] who added chia seeds (2, 4 and 6%) as fat replacers to chicken burgers.

### 3.2. Lipid Oxidation

Lipid oxidation can have a negative effect on the quality of meat and meat products as it usually modifies sensory attributes such as colour, taste or flavour, and changes the nutritional composition [16]. TBARS values in the analysed samples were ranging between 0.26–0.45 mg MDA/kg product (Table 1), which are distinctly lower than the level of incipient rancidity (>1.0) [25]. According to Grigioni et al. [26] there is a low possibility that consumers will be able to detect these off-flavours in meat product when TBA values are below 0.5 mg MDA/kg. TBARS values were influenced by storage time and their significant (*p* < 0.05) increase was noted after 6 days of storage in both samples, but in a greater extent in the control samples. During storage the pork patties with chia seeds were characterized by significantly lower (*p* < 0.05) TBARS values than the control samples. The antioxidant activity of chia seeds results from their bioactive compounds content such as tocopherols, polyphenols and carotenoids, which may be responsible for inhibition of lipid oxidation [27,28]. Antioxidant potential of chia seeds was reported by Coelho and Salas-Mellado [29], Segura-Campos et al. [30] and Ding et al. [31]. Similar observations were demonstrated by Fernández-López et al. [23] during refrigerated storage of frankfurters with chia products. According to Pintado et al. [25] the low TBARS level in frankfurters with chia flour, despite that it was higher than in control samples, generally suggest that oxidative changes were small and these products were of good quality.

### 3.3. Colour Parameters

Colour of the cross–section of pork patties was affected by chia seeds addition (Table 2). The samples with chia were characterized by a more intense red colour and less intense yellow colour at initial time of the experiment. On days 6 and 12 storage, significant differences between samples were noted only in parameter b*. During storage, the intensity of redness on the cross-section decreased in the samples with chia seeds addition and intensity of yellowness in the control samples. Similar results were reported by Paula et al. [24] who observed that chicken burgers, prepared with chia seeds, were characterized by lower values of L* and b* parameters and higher value of a* parameter than control samples. In our research, the samples with chia seeds showed lower hue angle values and were closer related to a pink colour than the control samples. In contrast, chroma (C*) described as the “saturation index”, refers to the intensity of the colour, and in the present research, the samples with chia seeds showed lower intensity of the colour than the control samples.

A significant effect of the addition of chia seeds on most parameters of the surface colour of pork patties was demonstrated (Table 3). At day 0 patties with chia seeds showed lower (*p* < 0.05) a*, b* and C* values of colour than the control samples. The hue angle in the samples with chia seeds was similar to the control samples. During storage, at days 6 and 12, significant differences were noted in L*, b* and C* values. The storage time had a significant effect only on the lightness of the colour and during storage values of this parameter significantly increased (*p* < 0.05) only in the control samples.

### 3.4. Textural Properties

Table 4 shows the textural parameters of analysed pork patties during storage. Lower (*p* < 0.05) values of hardness in whole period of storage were measured in the samples with chia seeds.

Hardness of the control samples has not changed during 12 days of storage, whereas the value of this parameter significantly increased after 6 days of storage in the patties with chia seeds and remained at similar level until the end of experiment. The pork patties with chia addition showed lower values of chewiness than the control samples after 0 and 12 days of storage. Time of storage had a significant (*p* < 0.05) effect on the value of this parameter only in the chia seeds samples and after 6 days of storage this parameter significantly increased and remained at the similar level after 12 days of storage. The patties with the addition of chia seeds were characterized by a similar cohesiveness to the control samples and there was no effect of the storage time on the value of this parameter. In terms of springiness, samples differed significantly only at day 0 and during storage this parameter significantly changed in the control samples but these changes were not unambiguous. The lower hardness and chewiness values can be perceived positively by consumers because these changes can be associated with a better eating quality of product. The results obtained in the present study confirm those reported by Fernández-López et al. [23] and Lucas-González et al. [16]. These authors showed that the addition of chia seeds and chia flour to frankfurters and the addition of chia oil emulsion to pork burgers, respectively, improved the tenderness and chewiness of those products. Different results were reported by Herrero et al. [32]. The authors found that frankfurters prepared with addition of chia flour were characterized by higher hardness, lower chewiness and lower elasticity than control samples.

### 3.5. Sensory Quality

The results of sensory evaluation are presented in Table 5. The products with chia seeds addition received distinctly lower scores for desirability of appearance than the controls at day 0, but the difference was not significant. During storage, the scores for this attribute were similar in both products. The addition of chia seeds was noticeable in patties at not very high aroma intensity level and similar intensity was noted at individual stages of storage.

The patties with chia seeds at days 0 and 6 of storage were characterized by similar juiciness and tenderness to the control samples, and after 12 days of storage they received higher scores for these attributes. Intensity of chia flavour and aroma were detectable at a similar level at the storage time considered and did not lower the acceptability scores for these attributes. The scores for overall quality of the patties were high (7.8–8.7 points) and the differences between the products were found only after 6 days of storage and higher scores were obtained by the product with chia seeds than the control sample. The effect of storage time on the overall quality of patties was found only in products with the addition of chia seeds and higher scores for this attribute were obtained for samples after 6 and 12 days of storage than at day 0.

Pintado et al. [22] did not show significant differences in colour, flavour, texture and general acceptability of frankfurters with emulsion gels containing 5% chia flour used as animal fat replacer. In another research Pintado et al. [33] showed that pork sausages with addition of emulsion gel containing 20% chia flour received lower marks for colour, texture and flavour than samples with animal fat and were judged worse for general acceptability, but nevertheless they were judged as acceptable by the panellists. Research by Ding et al. [31] showed that chia seeds improved sensorial properties of restructurized ham-like products. Fernández-López et al. [23] showed that 3% addition of chia products (chia seeds, chia flour and chia co-product) to frankfurters affected their sensory attributes such as colour, flavour, taste and general acceptability, but samples were judged as acceptable.

## 4. Conclusions

The present research suggests that addition of chia seeds gel to the meat product is a good option due to the possibility of improving the nutritional composition and pro-healthy properties in terms of fat content without negative effect on the technological properties of the final product. The patties with chia seeds gel showed higher intensity of red colour and lower intensity of yellow colour on the cross section as well as lower angle than the control samples, but colour was less saturated. Addition of chia seeds gel made the patties less tough and chewy which may be interesting for the consumers. Chia seeds gel did not change distinctly the overall quality of products with their addition and these products were evaluated as juicier and softer than the control samples. Additionally, the chia seeds addition slowed down lipid oxidation in product during storage.

## Figures and Tables

**Table 1 foods-10-01744-t001:** Effect of chia seeds and storage time on the water activity, pH and TBARS of pork patties (mean ± SD).

Parameters	Samples	Storage Time [Days]		
		0	6	12
Moisture [%]	control	69.15 ^A^ ± 0.12	ND	ND
	with chia seeds	70.63 ^B^ ± 0.13	ND	ND
Protein [%]	control	20.27 ^B^ ± 0.49	ND	ND
	with chia seeds	18.10 ^A^ ± 0.32	ND	ND
Fat [%]	control	9.08 ^B^ ± 0.87	ND	ND
	with chia seeds	8.47 ^A^ ± 0.76	ND	ND
Water activity [-]	control	0.981 ^aA^ ± 0.001	0.982 ^aA^ ± 0.002	0.982 ^aA^ ± 0.001
	with chia seeds	0.984 ^aA^ ± 0.002	0.985 ^aA^ ± 0.001	0.984 ^aA^ ± 0.001
pH [-]	control	6.55 ^bB^ ± 0.02	6.46 ^aB^ ± 0.02	6.47 ^aA^ ± 0.01
	with chia seeds	6.45 ^abA^ ± 0.03	6.35 ^aA^ ± 0.05	6.48 ^bA^ ± 0.04
TBARS [mg MDA/kg]	control	0.26 ^aA^ ± 0.01	0.42 ^bB^ ± 0.01	0.45 ^bB^ ± 0.02
	with chia seeds	0.27 ^aA^ ± 0.01	0.34 ^bA^ ± 0.01	0.35 ^bA^ ± 0.01

^a, b^—mean values in rows with different letter differ significantly at *p* < 0.05; ^A, B^—mean values in columns with different letters differ significantly at *p* < 0.05; ND—not determined.

**Table 2 foods-10-01744-t002:** Effect of chia seeds and storage time on the colour parameters on the cross section of pork patties (mean ± SD).

Parameters	Samples	Storage Time [Days]		
		0	6	12
L* [-]	control	65.85 ^aA^ ± 2.27	65.49 ^aA^ ± 2.89	65.92 ^aA^ ± 1.88
	with chia seeds	65.03 ^aA^ ± 2.43	64.75 ^aA^ ± 2.57	66.98 ^Aa^ ± 2.32
a* [-]	control	8.58 ^aA^ ± 0.89	9.12 ^aA^ ± 1.09	8.85 ^aA^ ± 0.77
	with chia seeds	9.78 ^aB^ ± 0.45	8.93 ^abA^ ± 0.67	8.43 ^bA^ ± 0.57
b* [-]	control	17.35 ^bB^ ± 0.89	16.69 ^abB^ ± 1.28	16.03 ^aB^ ± 1.05
	with chia seeds	14.92 ^aA^ ± 0.81	13.79 ^aA^ ± 0.80	13.84 ^aA^ ± 0.81
C*	control	19.37 ^aB^ ± 0.98	19.03 ^aB^ ± 0.55	18.32 ^aB^ ± 1.09
	with chia seeds	17.86 ^aA^ ± 1.04	16.44 ^aA^ ± 0.84	16.21 ^aA^ ± 1.06
h°	control	63.71 ^bB^ ± 2.38	61.41 ^aB^ ± 1.92	61.07 ^aB^ ± 2.17
	with chia seeds	56.83 ^aA^ ± 2.67	57.06 ^aA^ ± 2.00	58.71 ^aA^ ± 1.85

^a, b^—mean values in rows with different letter differ significantly at *p* < 0.05; ^A, B^—mean values in columns with different letter differ significantly at *p* < 0.05.

**Table 3 foods-10-01744-t003:** Effect of chia seeds and storage time on the colour parameters on surface of pork patties (mean ± SD).

Parameters	Samples	Storage Time [Days]		
		0	6	12
L* [-]	control	64.93 ^aA^ ± 1.10	66.90 ^abB^ ± 1.91	67.53 ^bB^ ± 1.86
	with chia seeds	65.61 ^aA^ ± 2.06	64.04 ^aA^ ± 2.08	64.99 ^aA^ ± 0.82
a* [-]	control	4.14 ^aB^ ± 0.43	3.77 ^aA^ ± 0.55	3.41 ^aA^ ± 0.52
	with chia seeds	3.49 ^aA^ ± 0.45	3.57 ^aA^ ± 0.67	3.27 ^aA^ ± 0.57
b* [-]	control	18.08 ^aB^ ± 0.62	18.03 ^aB^ ± 1.59	17.19 ^aB^ ± 1.14
	with chia seeds	14.97 ^aA^ ± 0.32	15.77 ^aA^ ± 1.52	15.33 ^aA^ ± 1.08
C*	control	18.56 ^aB^ ± 0.53	18.52 ^aB^ ± 1.66	17.53 ^aB^ ± 1.12
	with chia seeds	15.38 ^aA^ ± 0.32	16.18 ^aA^ ± 1.55	15.68 ^aA^ ± 1.17
h°	control	77.09 ^aA^ ± 1.65	78.3 1^aA^ ± 0.80	78.74 ^aA^ ± 1.79
	with chia seeds	76.90 ^aA^ ± 1.68	77.26 ^aA^ ± 2.10	78.01 ^aA^ ± 1.53

^a, b^—mean values in rows with different letter differ significantly at *p* < 0.05; ^A, B^—mean values in columns with different letter differ significantly at *p* < 0.05.

**Table 4 foods-10-01744-t004:** Effect of chia seeds and storage time on the texture parameters of pork patties (mean ± SD).

Parameters	Samples	Storage Time [Days]		
		0	6	12
Hardness [N]	control	5.45 ^aB^ ± 0.87	5.26 ^aB^ ± 0.71	5.43 ^aB^ ± 0.63
	with chia seeds	3.74 ^aA^ ± 0.55	4.98 ^bA^ ± 0.61	4.63 ^bA^ ± 0.61
Chewiness [N]	control	1.41 ^aB^ ± 0.42	1.18 ^aA^ ± 0.28	1.36 ^aB^ ± 0.37
	with chia seeds	0.82 ^aA^ ± 0.23	1.25 ^bA^ ± 0.42	1.12 ^bA^ ± 0.25
Cohesiveness [-]	control	0.39 ^aA^ ± 0.03	0.38 ^aA^ ± 0.03	0.38 ^aA^ ± 0.02
	with chia seeds	0.38 ^aA^ ± 0.02	0.38 ^aA^ ± 0.02	0.39 ^aA^ ± 0.02
Springiness [-]	control	0.64 ^bB^ ± 0.04	0.58 ^aA^ ± 0.04	0.63 ^bA^ ± 0.04
	with chia seeds	0.58 ^aA^ ± 0.05	0.59 ^aA^ ± 0.04	0.62 ^aA^ ± 0.04

^a, b^—mean values in rows with different letter differ significantly at *p* < 0.05; ^A, B^—mean values in columns with different letter differ significantly at *p* < 0.05.

**Table 5 foods-10-01744-t005:** Effect of chia seeds and storage time on the sensory quality of pork patties (mean ± SD).

Attributes	Samples	Storage Time [Days]		
		0	6	12
Appearance 1—undesirable 10—very desirable	control	8.0 ^aA^ ± 1.3	7.4 ^aA^ ± 1.7	7.6 ^aA^ ± 1.4
	with chia seeds	6.9 ^aA^ ± 1.5	7.7 ^aA^ ± 1.2	8.0 ^aA^ ± 13
Intensity of chia aroma 1—not detectable 10—very strong	control	NE	NE	NE
	with chia seeds	1.8 ^a^ ± 0.9	2.0 ^a^ ± 1.1	2.6 ^a^ ± 1.1
Aroma acceptability 1—undesirable 10—very desirable	control	7.8 ^aA^ ± 2.0	8.0 ^aA^ ± 1.1	7.9 ^aA^ ± 1.4
	with chia seeds	7.5 ^aA^ ± 1.1	7.4 ^aA^ ± 1.2	8.2 ^aA^ ± 0.9
Juiciness 1—dry 10—very juicy	control	7.2 ^aA^ ± 0.9	7.3 ^aA^ ± 0.8	7.4 ^aA^ ± 1.0
	with chia seeds	7.5 ^aA^ ± 0.9	8.0 ^aA^ ± 0.8	8.5 ^aB^ ± 1.1
Tenderness 1—tough 10—very tender	control	7.9 ^aA^ ± 0.9	8.2 ^aA^ ± 0.6	8.0 ^aA^ ± 0.8
	with chia seeds	7.9 ^aA^ ± 1.1	8.3 ^aA^ ± 1.1	8.7 ^aB^ ± 0.5
Intensity of chia flavour 1—not detectable 10—very strong	Control with chia seeds	NE 3.1 ^a^ ± 0.7	NE 2.7 ^a^ ± 0.5	NE 3.2 ^a^ ± 0.6
Flavour acceptability 1—undesirable 10—very desirable	control	7.1 ^aA^ ± 1.7	8.0 ^aA^ ± 1.0	8.4 ^aA^ ± 1.1
	with chia seeds	7.1 ^aA^ ± 1.6	8.0 ^aA^ ± 1.2	8.4 ^aA^ ± 0.8
Overall quality 1—very low 10—very high	control	8.3 ^aA^ ± 1.3	8.0 ^aA^ ± 0.8	8.1 ^aA^ ± 1.1
	with chia seeds	7.8 ^aA^ ± 0.9	8.7 ^bB^ ± 0.5	8.7 ^bA^ ± 0.7

^a, b^—mean values in rows with different letter differ significantly at *p* < 0.05; ^A, B^—mean values in columns with different letter differ significantly at *p* < 0.05; NE—not evaluated.

## Data Availability

Not applicable.

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
