# Peer review of "Effect of the Addition of Chia Seed Gel as Egg Replacer and Storage Time on the Quality of Pork Patties"

_foods, 2021, doi:10.3390/foods10081744_

Round 1
Reviewer 1 Report
This paper evaluated the pork patties made with egg and chia seeds gel replacer. The main drawback of the paper is the sensory analysis conducted. More details need to be added to explain what the sensory analysis trial entailed, as well as, it needs to be explained if the participants were trained to evaluate the pork patties or not. Specific comments can be found below:
Line 58- Is the preliminary study published?
Line 73- Was the production of the patties replicated in independent production batches?
Line 81- I think “tree” should be “three”?
Line 122- Was the TPA based on standardized methods for burgers or meat products? If so, they should be cited.
Line 142- Was the sensory analysis approved by an ethics board?
Line 143- How long were the patties stored before they were warmed up for the sensory trial?
Line 144- How big was each big size wedge? Why was jelly included?
Line 151- How was their sensory sensitivity confirmed? Were they trained on how to evaluate patties based on the attributes listed below?
Line 153- What were the anchors of the scale?
Line 155- Were the panelists trained? If so, why did they evaluate the acceptability of the products? This should be completed using consumers. If not, then 12 participants are much too low to evaluate the acceptability of the food items.
Line 155- Did the panelists only evaluate the patties once?
Line 272- The scales used in the sensory analysis should be added as a footnote.
Author Response
Comments in the review have been corrected in the text of the manuscript and are marked in red.
Line 58 - Is the preliminary study published?
Preliminary studies have not been published because they were needed to specify the composition of a product that would be sensorially attractive.
Line 73 - Was the production of the patties replicated in independent production batches?
The production of the patties was replicated in independent three production batches and this information was completed in the Sample preparation section.
Line 81 - I think “tree” should be “three”?
This mistake was corrected.
Line 122 - Was the TPA based on standardized methods for burgers or meat products? If so, they should be cited.
TPA details were applied on the basis of the method described for pork burgers and the reference to TPA method was added in the Texture profile analysis section.
Line 142 - Was the sensory analysis approved by an ethics board?
To my knowledge Sensory analysis does not require the approval of the ethic board, because the evaluation was performed by internal trained sensory panel using a specific evaluation method.
Line 143 - How long were the patties stored before they were warmed up for the sensory trial?
Samples were warmed immediately after opening the package and this information was completed in the Sensory quality section.
Line 144 - How big was each big size wedge? Why was jelly included?
For sensory analysis pork patties were cut in half and this information was completed in Sensory quality section. Pork patties did not include jelly and this mistake was removed from Sensory quality section.
Line 151 - How was their sensory sensitivity confirmed? Were they trained on how to evaluate patties based on the attributes listed below?
Before sensory analysis the panellists were tested according to the ISO standard 8586:2012 and trained in two separated sessions using pork patties as reference material. The reference ISO standard was added to the list and an appropriate information included in the Sensory quality section.
Line 153 - What were the anchors of the scale?
The anchors of the scale are located in Table 5 and an appropriate information was completed in the section on Sensory quality.
Line 155 - Were the panelists trained? If so, why did they evaluate the acceptability of the products? This should be completed using consumers. If not, then 12 participants are much too low to evaluate the acceptability of the food items.
The assessment involved people trained in sensory analysis and not consumers as the subject of the evaluation were various attributes including the quality and intensity of smell, taste and overall quality of products.
Line 155 - Did the panelists only evaluate the patties once?
The panelists evaluated the patties in two separate sets during sensory session. This information was added to Sensory quality section.
Line 272 - The scales used in the sensory analysis should be added as a footnote.
In our opinion the scales of individual attributes presented in table instead as footnote contribute to better readability of the results.
Reviewer 2 Report
Interesting work. Some sentences are too long and quite confusing. Please clarify the sentences highlighted in yellow in the attached file.

Author Response
All comments in the review have been corrected in the text of the manuscript and are marked in red.
Reviewer 3 Report
I have reviewed the manuscript titled: Effect of the addition of chia seed gel as egg replacer and storage time on the quality of pork patties.
This article aims to evaluate the use of chia seeds gel act as egg replacer on the quality of pork patties and to analyze the comical composition, pH value, color and texture parameters, sensoria acceptance and lipid oxidation during the storage time of 12 days at refrigerator. The information of this work is useful and relevant and there is possibility of improving the pro-properties of the meat patty products and nutritional composition of the manuscript that could be adapted by meat processing industry especially for frankfurters, sausage, burgers, chicken nuggets without adversely sensorial shelf-life in the future. I think the manuscript needs minor revision. The article is not innovative, however, it contains original and interesting information for meat processing of patty product. Abstract is well written upon and the color and texture of patties are mentioned and evaluated. Introduction is well addressed including phytocompounds of chia seeds as antioxidants, dietary fiber, emulsifier, and water holding agent. The information of pork patties consumption in Poland or whole world was not introduced and why the chia seeds could form gels was explain well in this study Actually I don’t think the chia seeds could form gel.
Materials and methods were well described.
This article would be accepted if the authors revised a few mistyping in Materials and method section and Table 1 as attached file.
I am not a native English speaker. The manuscript seems have no major mistakes are detected and the manuscript can be easily understood. The results are well discussed.
References
All references follow the required format for Foods. Therefore, it is acceptable after minor revision as attached file.
I enjoyed reading this manuscript; the needs of special groups of meat processing of meat patties. This manuscript presents some interesting and useful data.
Date of this review
9 July 2021 9:12

Author Response

(The authors gave the same response as above.)

Round 2
Reviewer 1 Report
Thank you to the authors; all of my concerns have been addressed.